# Resveratrol Decreases the Invasion Potential of Gastric Cancer Cells

**DOI:** 10.3390/molecules27103047

**Published:** 2022-05-10

**Authors:** Daniel Rojo, Alejandro Madrid, Sebastián San Martín, Mario Párraga, Maria Aparecida Silva Pinhal, Joan Villena, Manuel Valenzuela-Valderrama

**Affiliations:** 1Magister en Ciencias Médicas Mención Biología Celular y Molecular, Centro de Investigaciones Biomédicas, Escuela de Medicina, Universidad de Valparaíso, Angamos 655, Reñaca, Viña del Mar 2340000, Chile; daniel.rojo@postgrado.uv.cl (D.R.); sebastian.sanmartin@uv.cl (S.S.M.); mario.parraga@uv.cl (M.P.); 2Laboratorio de Productos Naturales y Síntesis Orgánica (LPNSO), Departamento de Química, Facultad de Ciencias Naturales y Exactas, Universidad de Playa Ancha, Avda. Leopoldo Carvallo 270, Playa Ancha, Valparaíso 2340000, Chile; alejandro.madrid@upla.cl; 3Departamento de Bioquímica, Universidade Federal de São Paulo (UNIFESP), São Paulo 04044-020, Brazil; cida.pinhal@fmabc.br; 4Laboratorio de Microbiología Celular, Instituto de Investigación y Postgrado, Universidad Central de Chile, Lord Cochrane 418, Santiago 8330546, Chile

**Keywords:** resveratrol, migration, invasion, NF-κB, SOD, heparanase, gastric cancer

## Abstract

The cancer-preventive agent Resveratrol (RSV) [3,5,4′-trihydroxytrans-stilbene] is a widely recognized antioxidant molecule with antitumoral potential against several types of cancers, including prostate, hepatic, breast, skin, colorectal, and pancreatic. Herein, we studied the effect of RSV on the cell viability and invasion potential of gastric cancer cells. AGS and MKN45 cells were treated with different doses of RSV (0–200 μM) for 24 h. Cell viability was determined using the Sulphorhodamine B dye (SRB) assay. For invasion assays, gastric cells were pre-treated with RSV (5–25 μM) for 24 h and then seeded in a Transwell chamber with coating Matrigel. The results obtained showed that RSV inhibited invasion potential in both cell lines. Moreover, to elucidate the mechanism implicated in this process, we analyzed the effects of RSV on SOD, heparanase, and NF-κB transcriptional activity. The results indicated that RSV increased SOD activity in a dose-dependent manner. Conversely, RSV significantly reduced the DNA-binding activity of NF-κB and the enzymatic activity of heparanase in similar conditions, which was determined using ELISA-like assays. In summary, these results show that RSV increases SOD activity but decreases NF-kB transcriptional activity and heparanase enzymatic activity, which correlates with the attenuation of invasion potential in gastric cancer cells. To our knowledge, no previous study has described the effect of RSV on heparanase activity. This article proposes that heparanase could be a key effector in the invasive events occurring during gastric cancer metastasis.

## 1. Introduction

Cancer is the second leading cause of death after myocardial infarction worldwide [1]. Gastric cancer is a neoplasm responsible for high mortality rates in developed and underdeveloped countries due to ineffective therapeutic approaches, including surgical resection of the stomach and chemotherapeutic regimens, resulting in relapse, metastasis, and poor survival [1]. Cancer cells can migrate from their primary sites and invade neighboring normal tissues; this process allows cancer cells to reach distant organs, a mechanism named metastasis [2,3]. Despite metastasis being the critical cause of cancer therapy failure and morbidity in affected patients [2], no therapy has been developed that successfully targets metastasis-associated processes of any human cancer type [4].

The degradation of basement membrane and extracellular matrix (ECM) structures are essential features of invasion and metastasis in gastric cancer. In this regard, heparanase activity plays a decisive role in tumor cell dissemination associated with the metastatic process and promotes tumor growth, angiogenesis, and metastasis [5,6,7,8]. Additionally, several matrix metalloproteinases (MMPs) participate in the proteolytic degradation of the ECM during the cancer invasion processes [2,9]. On the other hand, reactive oxygen species (ROS) are involved in the genesis of various cancer types by promoting oxidative DNA damage and deregulating gene expression [10,11,12]. Additionally, ROS can increase the invasive potential of malignant cells by inducing metalloproteinase gene expression [13,14].

Antioxidant enzymes are crucial to protect normal cells from oxidative stress damage. According to this view, dysregulation, or defects in the activity of antioxidant enzymes, such as Superoxide dismutase (SOD), Catalase (CAT), and Glutathione-peroxidase (GPX), are frequently associated with pathological processes [10]. For instance, the reduced activity or expression of antioxidant enzymes is related to developing several types of cancers [12,15]. Thus, current studies focus on the inhibitory effect of various antioxidant molecules on the invasive potential of cancer cells [16]. One example is the preventive cancer agent Resveratrol (RSV) [3,5,4′-trihydroxytrans- stilbene], a widely recognized antioxidant molecule with a demonstrated anticancer potential against several types of cancers, including prostate, hepatic, breast, skin, colorectal, and pancreatic cancer [17,18,19]. The reported effect of RSV against malignant cells in vitro and in vivo has been ascribed to its antioxidant and anti-inflammatory activities [20,21,22], which interferes with signal transduction pathways, for instance, by increasing antioxidant mediators such as Catalase and Superoxide dismutase, inhibiting angiogenesis, modulating cell cycle regulators, and inducing selective apoptosis [23,24,25,26,27]. According to this notion, the overexpression of extracellular SOD attenuates *HPSE* transcription and inhibits breast carcinoma cell growth and invasion [24].

RSV inhibits the nuclear factor-kappa B (NF-κB) signaling pathway [28,29], which has an important role in gastric cancer development [30]. NF-κB/Rel is a family of transcription factors that form homo or heterodimers between p50-, p52-, RelA/p65-, c-Rel-, and RelB-related proteins that are responsible for regulating the expression of inflammatory cytokines, chemokines, immunoreceptors, antiapoptotic proteins, cell adhesion molecules, and invasion/metastasis-related molecules, among others [31]. In the canonical pathway of NF-κB activation, the membrane receptors promote the phosphorylation and degradation of IκBα by IKKα/IKKβ kinases, which sequester p65/p50 heterodimers in the cytoplasm. These events lead p65/p50 to translocate to the nucleus [31,32,33,34]. NF-κB activity promotes gastric cancer development; however, its role is unclear during gastric cancer metastasis. Of note, NF-κB is a transcriptional activator of *MMP-9* and urokinase-type plasminogen activator (*uPA*) genes [35,36], which are highly upregulated during metastasis, supporting the role of NF-κB signaling in the metastatic process [37]. Additionally, it was reported that the nuclear accumulation of NF-κB subunits correlated with increased heparanase expression and poor clinicopathological features of gastric cancer specimens [38], suggesting an important role for NF-κB signaling in the gastric metastatic process. Considering these antecedents, we hypothesize that RSV might affect the invasive potential of gastric cancer cells by modulating NF-κB, SOD, and heparanase activity.

## 2. Results

### 2.1. RSV Treatment Reduced Cell Viability of Gastric Cancer Cells

As outlined above, RSV treatment reduces gastric cancer cell viability [39]; however, we wondered if RSV could affect other processes associated with cancer progression beyond this reported cytotoxic effect. With this purpose in mind, we assessed a wide range of RSV concentrations to determine those doses that had minimal impact on cell viability. Then, we exposed the human gastric cancer cell lines AGS and MKN45 to increasing concentrations of RSV for 24 h and analyzed the changes in the cell viability using the SRB dye assay (see Section 4). As shown in Figure 1, gastric cells displayed significant sensitivity to RSV treatment, which inhibited viability in a dose-dependent manner in both cell lines. Of note, the higher doses of RSV (25–200 μM) significantly reduced the cell viability in both AGS and MKN45 cell lines. Conversely, lower concentrations of RSV (5–12.5 µM) did not show statistically significant growth inhibition.

### 2.2. RSV Reduced the Invasion Potential of Gastric Cancer Cells

Next, we determined the effect of RSV on the invasion potential of gastric cancer cells by Boyden’s chamber approach [40]. Considering the results obtained using the SRB dye assay (see Figure 1), we used lower concentrations of RSV (5 and 12.5 μM) to avoid a significant cytotoxic effect; however, we also included an effective cytotoxic concentration (RSV 25 µM) to have a comparative point. As shown in Figure 2, RSV reduced the invasion potential in both AGS and MKN45 cell lines in a dose-dependent manner. Of note, RSV (5–25 μM) treatment significantly reduced the invasion potential of AGS cells between 60 and nearly 90% compared to the control condition. On the other hand, RSV treatment significantly diminished the invasion of MKN45 between 75 and nearly 95% compared to the control condition. Interestingly, RSV 25 µM reduced invasion considerably (90 and 95% in AGS and MKN45, respectively) compared to similar concentrations with less effect on cell viability (45 and 40% in AGS and MKN45, respectively). According to these results, it is possible to conclude that RSV can affect different processes associated with carcinogenesis depending on the concentration used. This potent inhibitory effect agrees with data obtained in similar in vitro studies using cancer cell lines from other origins [36,41,42]. Here, we confirm that non-cytotoxic concentrations of RSV significantly reduce gastric cancer cell invasion.

### 2.3. Evaluation of SOD, Heparanase, and NF-κB Transcription Activity in Gastric Cancer Cells Treated with RSV

To elucidate the possible underlying mechanism by which RSV inhibits the invasion potential of the gastric cancer cells, we analyzed the changes in SOD activity in AGS and MKN45 cells following RSV treatment with an enzymatic approach (see Section 4). As shown in Figure 3, RSV treatment increased SOD activity after 24 h in a dose-dependent manner compared to the control condition in both cell lines; however, this increase was statistically significant only when gastric cancer cells were treated with the highest RSV concentration used in this assay (*p* < 0.05). In a previous study, augmented SOD activity was related to the diminished expression and activity of heparanase, resulting in decreased invasion ability in breast cancer cells [24]; thus, our experimental model confirms such findings.

On the other hand, it was reported that NF-κB partially regulated the expression of *HPSE* in the gastric cancer cell line MKN74 [43]. With this in mind, we explored the effect of RSV on the DNA-binding activity of NF-κB, with AGS and MKN45 cells treated with different RSV concentrations (5–25 μM) and ethanol 1% or the NF-κB inhibitor, Gliotoxin (100 ng/mL), as controls. After 24 h of treatment, nuclear fractions were collected, and NF-κB DNA-binding activity in nuclear extracts was determined using an ELISA-like assay (Materials and Methods). As shown in Figure 4, RSV 5–25 µM treatment significantly reduced the DNA-binding activity of NF-κB by nearly 50−70% in both cell lines (*p* < 0.05).

These results suggest that the inhibitory effect of RSV treatment on invasion potential is also associated with diminished NF-κB transcriptional activity.

Given that NF-κB activation has been linked to increased *HPSE* expression [38], and our results show that RSV reduced NF-κB transcriptional activity in our experimental model, we analyzed the changes in heparanase activity in gastric cancer cells following RSV treatment following the degradation of a biotinylated-HS substrate (see Section 4). As shown in Figure 5, heparanase activity was reduced in AGS cells treated with 5–25 µM of RSV for 24 h; however, this reduction was only statistically significant when AGS cells were treated with 5 and 12.5 µM of RSV compared with ethanol-1%-treated cells (100% activity). On the other hand, although it was possible to observe a decrease in heparanase activity in response to RSV 5 and 12.5 µM in MKN45 cells, such a reduction was not significant.

These results show that RSV reduces heparanase activity in gastric cancer cells, which correlates with increased SOD activity and NF-κB transcriptional activity inhibition. Heparanase is a critical effector during the metastasis process in gastric cancer development by promoting the invasion of malignant cells to distant organs. Importantly, this study provides the first evidence that RSV treatment significantly reduces the heparanase activity in gastric cancer cells.

## 3. Discussion

RSV has demonstrated antitumor activity in vitro and in vivo [18,22]. Thus, we evaluated a wide range of RSV concentrations to determine which could affect different processes relevant to cancer development, particularly the invasion potential of gastric cancer cells. Our study shows that RSV reduces cell viability and the invasion potential of two gastric cancer cell lines in a dose-dependent manner but at different concentrations. In keeping with our results, similar studies have shown that RSV can inhibit proliferation in other gastric cancer cells (KATO-III) and induces apoptosis and S/G2 cell cycle arrest in various cancer cell types in vitro [39,44]. The results obtained here reveal that RSV decreases cell viability in a dose-dependent fashion, becoming cytotoxic at higher concentrations (25–200 μM). However, at non-cytotoxic concentrations, it inhibits the invasion potential of AGS and MKN45 gastric cells. Based on these findings, we could hypothesize that the doses of RSV used in a possible therapy could be optimized to promote two different types of antitumor effects. On the one hand, high concentrations of RSV could trigger the death of malignant cells and, on the other, low doses could attenuate the processes involved in metastasis. This point is relevant given that some adverse hormetic effects have been reported during in vivo experiments using high doses of RSV [45].

One crucial feature of metastasis is the increased ability of tumor cells to migrate and invade distant tissues. Thus, metastasis is accompanied by various physiological alterations, including exacerbated ECM degradation resulting from the induced proteolytic activity involving MMPs and heparanase activities [2,9]. These enzymes degrade the ECM components—such as heparan sulfate proteoglycans, collagens, fibronectins, and laminins—allowing cancer cells to migrate and invade, thus offering a potential therapeutic target for complementary therapies [46,47]. Our results demonstrate that RSV reduced the invasion potential of AGS and MKN45 cells in vitro. Besides the effect of RSV on cell viability at high concentrations, at low concentrations, RSV 25 µM almost suppressed invasion completely in both cell lines and RSV 5 µM diminished by more than a half. These results are similar to those obtained using a siRNA against heparanase mRNA in gastric cancer cells [48]. Additionally, a natural RSV analog, Piceatannol, displayed similar inhibitory effects on invasion potential in breast cancer cells in vitro [49]. In this regard, our results showed that RSV reduces heparanase activity in the gastric cancer cells used in this study.

Previous works using the gastric cell line MKN74 showed that NF-κB partially regulates the expression of *HPSE* since, by inhibiting its transcriptional activity, heparanase expression is reduced [43]. Additionally, heparanase expression is positively related to advanced tumor stages (TNM classification), invasion depth, and poor prognosis in gastric cancer [50]. On the other hand, RSV also decreased NF-κB activity. NF-κB activation has been linked to the increased expression of heparanase during the invasion of gastric cancer cells [38]. Thus, we wondered if the NF-κB transcriptional activity could be modulated by RSV treatment in gastric cancer cell lines. In our experimental model, approximately a 50% decrease in NF-κB activation was observed when treating the AGS cell line with RSV at 25 μM and 12.5 μM. In addition, a decrease of about 50 to 70% in the activation of NF-κB was found according to the dose of RSV in the MKN45 cell line. Decreased activity in both cell lines presented statistical significance (*p* < 0.05). The inhibition of NF-κB in the presence of RSV was contrasted with Gliotoxin (a specific inhibitor of NF-κB, 100 ng/mL) in both cell lines. The results showed that RSV has a comparable effect to Gliotoxin in AGS and MKN45 cell lines, reducing NF-κB transcriptional activity.

ROS production has been implicated in the pathological induction of MMPs, heparanase, and uPA during the invasion of malignant cells [51]. In agreement with this observation, ROS sequestration, as a consequence of SOD overexpression, was related to the diminished expression and activity of heparanase in breast cancer cells, resulting in decreased invasion ability [24]. Accordingly, heparanase inhibition was correlated with the increased activity of SOD in gastric cancer cells treated with RSV. These results suggest that RSV can increase SOD activity and inhibit NF-κB signaling, decreasing heparanase activity and subsequent tumor invasion and metastasis.

Heparanase activity was evaluated to determine the impact of RSV on this proteolytic enzyme. Our results suggest that heparanase is a relevant effector during the invasion of gastric cancer cells and that its activity would be affected by the effect of RSV on SOD and NF-κB activities. Notably, non-toxic RSV concentrations (5–12.5 µM) decreased heparanase activity in both cell lines, although it was only statistically significant in AGS cells. These results provide valuable evidence of a mechanism by which RSV reduces the invasive capacity of the AGS and MKN45 cell lines. That evidence derives from the fact that RSV has a biological effect on SOD and NF-κB. Although evidence was found regarding heparanase as a critical enzyme in the invasive process, it is known that RSV affects the activity and levels of MMPs, especially MMP-2 and MMP-9 [52]. RSV diminishes MMP-9 expression through NF-κB inhibition. For that reason, the invasion results are probably due to RSV-mediated MMP-9 and heparanase inhibition. Our data indicate for the first time that RSV significantly decreases the heparanase activity in gastric cancer cell lines AGS and MKN45. To our knowledge, no study has described the effect of RSV on heparanase activity, although there are reports describing the impact of Resveratrol on SOD and NF-κB activities.

In addition, there is evidence that heparanase is involved in physiological processes of neovascularization, inflammation, and leukocyte migration, among others [52]. Recently it was proved possible to generate a mouse knockout to heparanase, which was phenotypically healthy, with life according to the average and fertile. Based on the finding of the increased expression of MMPs (especially MMP-2 and MMP-14) in heparanase-knockout, the authors propose that these enzymes would compensate for the decrease in heparinase activity [53].

## 4. Materials and Methods

### 4.1. Cell Culture and RSV Treatment

The human gastric cancer cells AGS and MKN45 were cultured in RPMI 1640 (Gibco, San Diego, CA, USA) supplemented with 10% (volume/volume) fetal bovine serum (FBS) at 37 °C in a humidified 5% CO_2_ incubator. For RSV treatment, appropriate volumes from a stock solution of RSV 1 mM (Sigma, St. Louis, MO, USA) in 100% ethanol were added to the cell medium to achieve the desired concentrations (5–200 μM), and then cells were incubated for the indicated periods.

### 4.2. Cell Viability Assay

Sulphorhodamine B (SRB) (Sigma, St. Louis, MO, USA) dye assay was performed to determine cell viability, as previously described [54]. Briefly, gastric cancer cell lines were seeded at the density of 3 × 10^3^ cells/well in 96-well plates. After 24 h of culture, cells were treated with different doses of RSV (ranging from 5 up to 200 μM) for 72 h. Then, cells were fixed, washed, and incubated with 50 µL of SRB solution (0.1% weight/volume in 1% acetic acid) per well and incubated at room temperature for an additional 30 min. Unbound SRB was removed by washing with 1% acetic acid. Plates were air-dried, and the retained SRB was solubilized with 100 µL of 10 mM Tris base (pH 10.5). Optical densities were read in a spectrophotometer plate reader at 540 nm. Values are shown as mean ± S.D., *n* = four independent experiments in triplicate.

### 4.3. Cell Invasion Assay

For cell invasion assays, a quantity of 2 × 10^5^ cells/well were seeded in a Transwell chamber with coating Matrigel (Sigma-Aldrich, St. Louis, MO, USA) as previously described [40]. Briefly, gastric cells were treated with RSV (5–25 μM) for 24 h or with Gliotoxin (100 ng/mL) for 1 h as a positive control and seeded into the upper chamber in a serum-free medium. A volume of 0.5 mL of medium containing 20% FBS was added to the lower chamber as a chemoattractant. After 48 h of incubation, the upper surface of the porous membrane was wiped off with a cotton swab. Cells that invaded the lower surface of the porous membrane were fixed with 50% methanol and stained with 0.1% crystal violet. Stained cells were counted in twenty random fields per filter, in a total of three filters (*n* = 3). Invasion was presented as percentage of Invasion = (number treated cells/number of control cells) × 100.

### 4.4. Determination of SOD Activity

For the determination of SOD activity, a number of 5 × 10^5^ cells/well were seeded in 6-well plates and cultured for 24 h. Next, cells were treated with RSV (5–25 μM) or ethanol 1% for 24 h. Before treatment, cells were collected, lysed, and centrifuged (14,000× *g* for 5 min). The supernatants were collected, and the SOD activity was measured using the SOD Activity ELISA-kit (Catalog #K335, BioVision, Waltham, MA, USA), according to the instructions provided by the manufacturer. Results were normalized to total protein content as described previously [55].

### 4.5. Determination of NF-κB DNA-Binding Activity

For the determination of NF-κB DNA-binding activity, a set of 5 × 10^5^ cells/well were seeded in 6-well plates and cultured for 24 h. Then, cells were treated with RSV (5–25 μM) or ethanol 1% for 24 h. Additionally, a negative control was included by treating the cells with Gliotoxin (100 ng/mL) for 2 h. Following treatment, cells were collected and centrifuged at 300× *g* for 5 min at 4 °C. Nuclear fractions were isolated and subsequently analyzed for NF-κB activity using the NF-κB (p50/p65) transcription factor assay kit (catalog #10011223, Cayman, Ann Arbor, MI, USA), following the manufacturer’s instructions.

### 4.6. Determination of Heparanase (HPSE) Activity

AGS and MKN45 gastric cells were treated with RSV (5–25 μM) or ethanol 1% for 24 h. Heparanase activity was measured using an ELISA-like assay using biotinylated heparan sulfate [56]. Briefly, the 96-well plate (Nalge Nunc, Rochester, NY, USA) was incubated with 200 µL of protamine sulfate (Sigma-Aldrich, St. Louis, MO, USA, 10 µg/mL) overnight at 37 °C. Subsequently, 100 µL of biotinylated heparan sulfate (10 µg/mL) was immobilized for 18 h at 37 °C. The enzymatic assay was performed in sodium acetate 25 mM, pH 5.5, in a final volume of 100 µL. A volume of 50 μL of the cellular extract was incubated with the pre-coated 96-well plate with biotinylated heparan sulfate. The immobilized biotinylated heparan sulfate not digested by heparanase was bound to streptavidin conjugated with europium. Then, an enhancement solution (200 μL) (PerkinElmer Life Sciences) was added to release the europium. Free europium was measured, and the data were analyzed in the MultiCalc software v2.7 (PerkinElmer Life Sciences-Wallac Oy, Turku, Finland). The product obtained via HPSE1 was expressed by the ratio of degraded heparan sulfate (HS) and total protein from the cellular fraction (μg degraded HS/μg total protein). The values were expressed as a percentage of heparanase activity.

### 4.7. Statistical Analysis

All assays were repeated in triplicate in at least three independent experiments, and all data were expressed as means ± S.D. Analysis of variance (ANOVA) for multiple comparisons was used as noted. In all cases, *p* < 0.05 was considered significant. All statistical tests were performed with the statistical analysis software Prism version 5 (GraphPad, San Diego, CA, USA)).

## 5. Conclusions

Resveratrol is a natural compound produced by several plants that has emerged as a potent antitumoral agent, whose mechanism of action involves its antioxidant and anti-inflammatory activities. In this work, we explored the ability of this molecule to inhibit the proliferation and invasion potential of gastric cancer cells. We showed that RSV reduces heparanase activity in AGS and MKN45 cells, which correlates with increased SOD activity and NF-κB transcriptional activity inhibition. Since heparanase is a critical effector during the metastasis process in gastric cancer development, these results provide valuable information for designing effective therapies based on the use of RSV. Currently, there are dozens of ongoing clinical trials with RSV (https://clinicaltrials.gov/ (accessed on 1 March 2022)) for both use and chemopreventive as a chemotherapeutic adjuvant, which seek to transfer to human medicine with favorable antineoplastic findings at the preclinical level. We hope that the results of this work and future preclinical studies in gastric cancer will lead to new clinical trials in order to get the most out of this promising molecule.

## Figures and Tables

**Figure 1 molecules-27-03047-f001:**
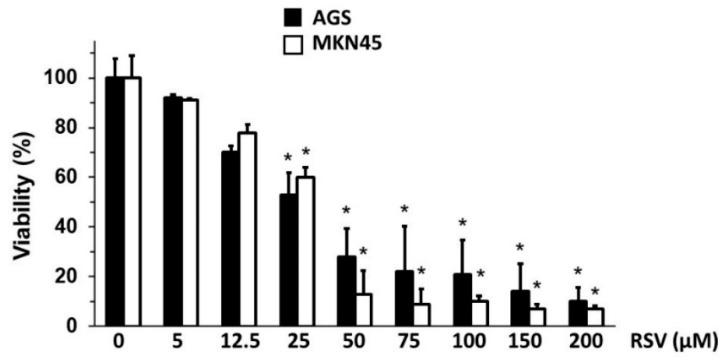
Effect of RSV treatment on cell viability of gastric cancer cell lines. AGS and MKN45 gastric cancer cells were treated with different concentrations of RSV (5–200 μM) for 24 h. Cell viability was determined using the SRB dye assay. Bars represent the percentage of viability in AGS (black bar) and MKN45 (white bar) cells in a dose-dependent manner compared to the control ethanol-treated cells (control 100% viability). Values represent means ± S.D. of four independent experiments. (* *p* < 0.05).

**Figure 2 molecules-27-03047-f002:**
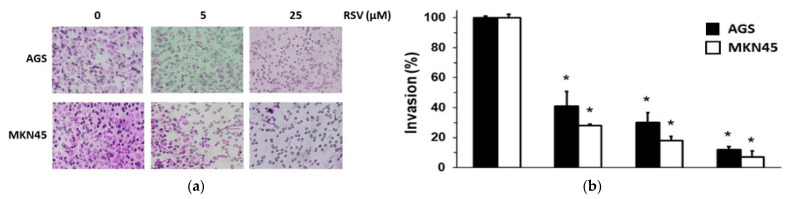
Effect of RSV on invasion potential of gastric cancer cell lines. AGS and MKN45 gastric cancer cells were treated with different concentrations of RSV (5–25 μM) for 24 h and then harvested and seeded into the upper compartment of the Boyden’s invasion chamber. The number of cells that moved into the lower chamber represented the invading cells. (**a**) After the RSV treatment, representative pictures of the invasion assay in AGS and MKN45 cells were presented (magnification ×100). (**b**) Bars represent the percentage of the invasion of AGS (black column) and MKN45 (white column) cells treated with different doses of RSV compared to the control ethanol-1%-treated cells (control 100% invasion). Data represent means ± S.D. of three independent experiments. * *p* < 0.001 vs. control.

**Figure 3 molecules-27-03047-f003:**
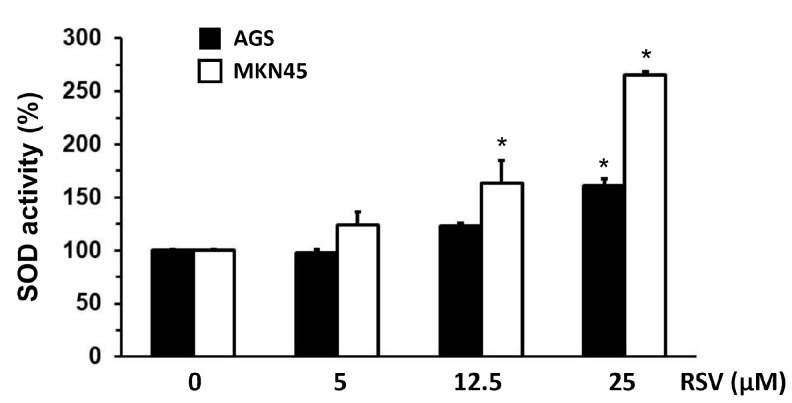
Effect of RSV on SOD activity in gastric cancer cell lines. AGS and MKN45 gastric cells were treated with different concentrations of RSV for 24 h (5–25 μM). SOD activity was determined using an ELISA method in AGS (black column) and MKN45 (white column) protein extracts. Data represent means ± S.D. of three independent experiments. * *p* < 0.05 vs. control.

**Figure 4 molecules-27-03047-f004:**
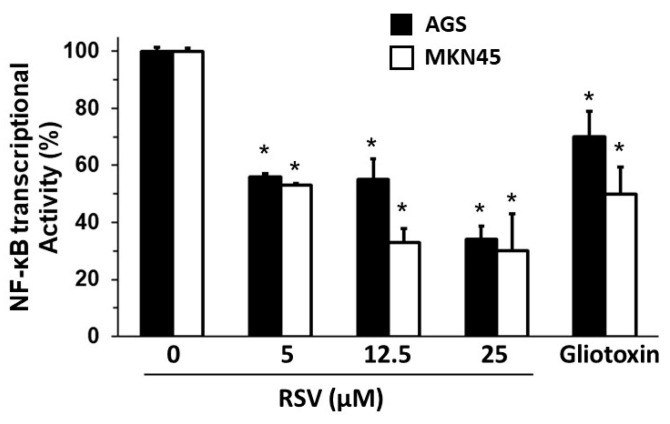
Effect of RSV on NF-κB DNA-binding activity of gastric cancer cell lines. AGS and MKN45 gastric cancer cells were treated with different concentrations of RSV (5–25 μM). Bars represent the percentage of NF-κB DNA-binding activity in nuclear extracts from AGS (black column) and MKN45 (white column) compared to the control ethanol-treated cells (control 100% activity). NF-κB inhibitor Gliotoxin (100 ng/mL) was included as a positive control. Data represent means ± S.D. of three independent experiments. * *p* < 0.05 vs. control.

**Figure 5 molecules-27-03047-f005:**
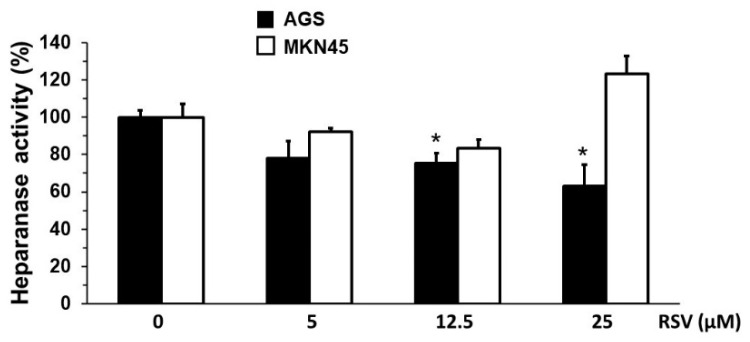
Effect of RSV on the heparanase activity of gastric cancer cell lines. AGS and MKN45 gastric cancer cells were treated with different concentrations of RSV (5, 12.5, and 25 μM) for 24 h. Heparanase activity was measured using biotinylated heparan sulfate as a substrate. Bars represent the percentage of enzymatic activity in AGS (black column) and MKN45 (white column) following RSV treatment compared to the ethanol-treated cells (control 100%). Values represent the means ± S.D. from triplicate experiments. * *p* < 0.05 vs. control.

## Data Availability

Not applicable.

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
