# Peer review of "Resveratrol Decreases the Invasion Potential of Gastric Cancer Cells"

_molecules, 2022, doi:10.3390/molecules27103047_

Round 1

Reviewer 1 Report

The paper describes the effects of the putative anti-cancer molecule Resveratrol (RSV) in different gastric cancer cell lines (AGS and MKN45). The authors focus their attention in: i) cell viability; ii) invasion potential; iii) evaluation of SOD, heparanase and NF-κB activity. This is a very interesting topic as cancer still causes a significant number of deaths worldwide. The research line is well defined and the main results are useful to a future translational scenario. Therefore, the present work is suitable for publication in Molecules.

I would like to suggest to the authors an extra point to be further explored/discussed. It was showed that “the higher doses of RSV (25-200 μM) significantly reduced the cell viability in both AGS and MKN45 cell lines. Conversely, lower concentrations of RSV (5-12.5 μM) did not show a statistically significant growth inhibition”. Nevertheless, in the remain experiments (invasion potential and evaluation of SOD/heparanase/NF-κB activity), lower concentrations of RSV (5-25 μM) were used “to avoid a significant cytotoxic effect”. Is there a contradiction here? If so, it can bring implications to the use of RSV as a viable anti-cancer agent? A brief discussion on these topics will add value to the interesting findings herein reported.

Other minor points to be checked/corrected:

Figure 1 (line 98) – The bars referring to the 200uM RSV concentration present the respective S.D. values similarly to the other tested concentration values?

Lines 250 and 258 – 3×103 and 2x105 cells/well should be replaced by 3×103 and 2×105 cells/well, respectively.

Line 288 – 37º should be replaced by 37ºC

Author Contributions, Funding and Acknowledgments sections (lines 311-322) – This sections should be properly revised removing the original journal text.  

Author Response

Response

Reviewer 2 Report

Manuscript Number: Molecules-1656501

Title: “Resveratrol decreased the invasion potential of gastric cancer 2 cells” by Manuel Valenzuela-Valderrama et al.

The authors have studied the effects of resveratrol on invasion potential against cancer. They have compared the sensitivities of different gastric cancer cells to resveratrol treatment through the sulphorhoda-mine B dye assay and they determined the cell viability. They reported that the resveratrol treatment diminished the Heparanase activity of gastric cancer cells significantly in all concentrations tested. Their data correlated with the RSV effects over SOD activity and NF-κB activation. They proposed that the Heparanase might be the effector in invasion events in gastric cancer.

The research work carried out by the authors is interesting and significant. The following are the comments which must be considered and to be addressed by the authors to improve the quality of the manuscript.

  1. The abstract: reframe the abstract with important findings of the work.
  2. “sulphorhoda-mine B” is it the same word or a split word?
  3. Try to include some more recent references in the introduction section.
  4. In section-2, the results and their significance need to be more elaborated.
  5. The result and analysis part needs a clearer explanation with significant findings.
  6. In line 243 what (v/v) mean?
  7. Provide the data set for all the assays in the supplementary material.
  8. Are the calculated values -S.D are within the acceptable range?
  9. Any comparative study with the reported results?
  10. Any structural studies or results obtained?
  11. The conclusion section must be redrawn with the significant findings of the work.

With all the above clear comments, I strongly recommend a major revision of the manuscript. After significant improvement, the manuscript can be considered for publication.

Author Response

response

Round 2

Reviewer 2 Report

Author's have revised the manuscript by considering the  reviewers comment. Now the manuscript is in the acceptable quality. It can be published in the journal Molecules.